# Seroprevalence of anti-SARS-CoV-2 antibodies before and after implementation of anti-COVID-19 vaccination among hospital staff in Bangui, Central African Republic

Alexandre Manirakiza[1,2]*, Christian Malaka[1], Hermione Dahlia Mossoro-Kpinde[2], Brice Martial Yambiyo[1,2], Christian Diamant Mossoro-Kpinde[2], Emmanuel Fandema[2], Christelle Niamathe Yakola[2], Rodrigue Doyama-Woza[2], Ida Maxime Kangale-Wando[2], Jess Elliot Kosh Komba[3], Sandra Manuella Bénedicte Nzapali Guiagassomon[2], Lydie Joella-Venus de la Grace Namsenei-Dankpea[1], Cathy Sandra Gomelle Coti-Reckoundji[1], Modeste Bouhouda[1], Jean-Chrisostome Gody[2,3], Gérard Grésenguet[2], Guy Vernet[1], Marie Astrid Vernet[1], Emmanuel Nakoune[1,2]

1 Institut Pasteur of Bangui, Pasteur International Network, Bangui, Central African Republic, 2 University of Bangui, Faculté des Sciences de la Santé, Bangui, Central African Republic, 3 Centre Hospitalier et Universitaire Pédiatrique de Bangui, Bangui, Central African Republic

* amanirak@yahoo.fr

## Abstract

Healthcare workers (HCWs) are at high to very high risk for SARS-CoV-2 infection. The persistence of this pandemic worldwide has instigated the need for an investigation of the level of prevention through immunization and vaccination against SARS-CoV-2 among HCWs. The objective of our study was to evaluate any changes in anti-COVID-19 serological status before and after the vaccination campaign of health personnel in the Central African Republic. We carried out a repeated cross-sectional serological study on HCWs at the university hospital centers of Bangui. Blood samples were collected and tested for anti-SARS-CoV-2 IgM and IgG using the ELISA technique on blood samples. A total of 179 and 141 HCWs were included in the first and second surveys, respectively. Of these staff, 31.8% of HCWs were positive for anti-SARS-CoV-2 IgG in the first survey, whereas 95.7% were positive for anti-SARS-CoV-2 IgG in the second survey. However, the proportion of HCWs positive for SARS-CoV-2 IgM antibodies was low (9.7% in the first survey and 3.6% in the second survey). These findings showed a sharp increase in seroprevalence over a one-year period. This increase is primarily due to the synergistic effect of the infection and the implementation of vaccines against COVID-19. Further studies to assess the persistence of anti-SARS-CoV-2 antibodies are needed.

## Introduction

Since the emergence of severe acute respiratory syndrome coronavirus 2 (SARS-CoV-2) in China in 2019, the world's population has been experiencing a viral pandemic [1]. On March

**Data Availability Statement:** An anonymized dataset can be downloaded from: https://doi.org/10.6084/m9.figshare.23694741.

**Funding:** This study was financially supported by the European Union via MEDILABSECURE (https://www.medilabsecure.com/) and the Institut Pasteur Network association. The funder of the study had no role in study design, data collection, data analysis, data interpretation, or writing of the report.

**Competing interests:** The authors have declared that no competing interests exist.

11, 2020, coronavirus disease 2019 (COVID-19) was declared a pandemic disease by the World Health Organization (WHO). The WHO had reported a total of 112,456,453 confirmed COVID-19 cases and 2,497,514 deaths (7.1% case fatality rate) worldwide as of February 2021 [2]. This COVID-19 pandemic has shaken up the working world, especially in healthcare facilities. Healthcare workers (HCWs) were quickly identified as being at risk of contracting COVID-19. HCWs work around the clock and are directly involved in the diagnosis, treatment, and care of COVID-19 patients and thus are at high risk of being infected with SARS-CoV-2. In the epidemiological setting of community transmission, HCWs are also at high to very high risk for SARS-CoV-2 infection [2–4].

Although the general population needs to stay home to reduce the spread of this virus, HCWs are doing the exact opposite. In some countries, HCWs work with inadequate protection and are at constant risk of contracting COVID-19. They must be constantly monitored because, if infected, they can readily transmit the virus to their colleagues, hospitalized patients, and even family members. Increasing infection rates among HCWs could lead to the collapse of the healthcare system and a further worsening of the pandemic: if there are too few staff, the situation would be even more difficult to manage [5].

WHO has prioritized the use of vaccines and HCWs are at the highest priority level. This level of priority was motivated by the need to protect these workers to ensure the availability of critical essential services in the response to the COVID-19 pandemic and to protect them from more severe forms of COVID-19. Further, health professionals and public health authorities play a central role in discussing COVID-19 vaccination with their patients [2, 6]. Epidemiological surveys can provide serological data to estimate the penetration of the virus in a given population, including the HCW population. Serological tests determine whether a person has produced antibodies in response to infection with the virus or vaccination [7].

In the Central African Republic (CAR), the Ministry of Health and Population was alerted in February 2020 and thereafter quickly implemented measures to fight the pandemic through building awareness, enhancing prevention and monitoring people traveling from high transmission areas and arriving in the CAR. On March 14, 2020, the first confirmed COVID-19-positive case was detected in the CAR. From that day until August 22, 2022, 14,803 cases have been confirmed, 14,520 patients have been cured, and 113 have died. The country has experienced four waves of this epidemic [8, 9].

In the CAR, anti-COVID-19 vaccination has been deployed since May 20, 2021. Currently, this pandemic persists worldwide with the emergence of SARS-CoV-2 variants [10]. Accurate identification of people who have previously had COVID-19 is important in measuring disease spread and assessing the success of public health interventions [11].

Given the context, the purpose of this study was to investigate the level of prevention through immunization and vaccination against SARS-CoV-2 among HCWs in university hospitals in Bangui, the capital of the CAR. The objective of our study was to evaluate any changes in the anti-COVID-19 serological status before and after the vaccination campaign of HCWs in the CAR.

## Method

This is a repeated cross-sectional study with descriptive and analytical purposes conducted at the university hospital centers of Bangui, capital of the CAR. The first survey was conducted in April 2021 at the Pediatric University Hospital of Bangui (CHUPB), the University Hospital of the Sino-Central African Friendship (CHUASC) and the Community University Hospital (CHUC). The second survey was conducted in May 2022 at the University Hospital of the Sino-Central African Friendship (CHUASC) and the Community University Hospital

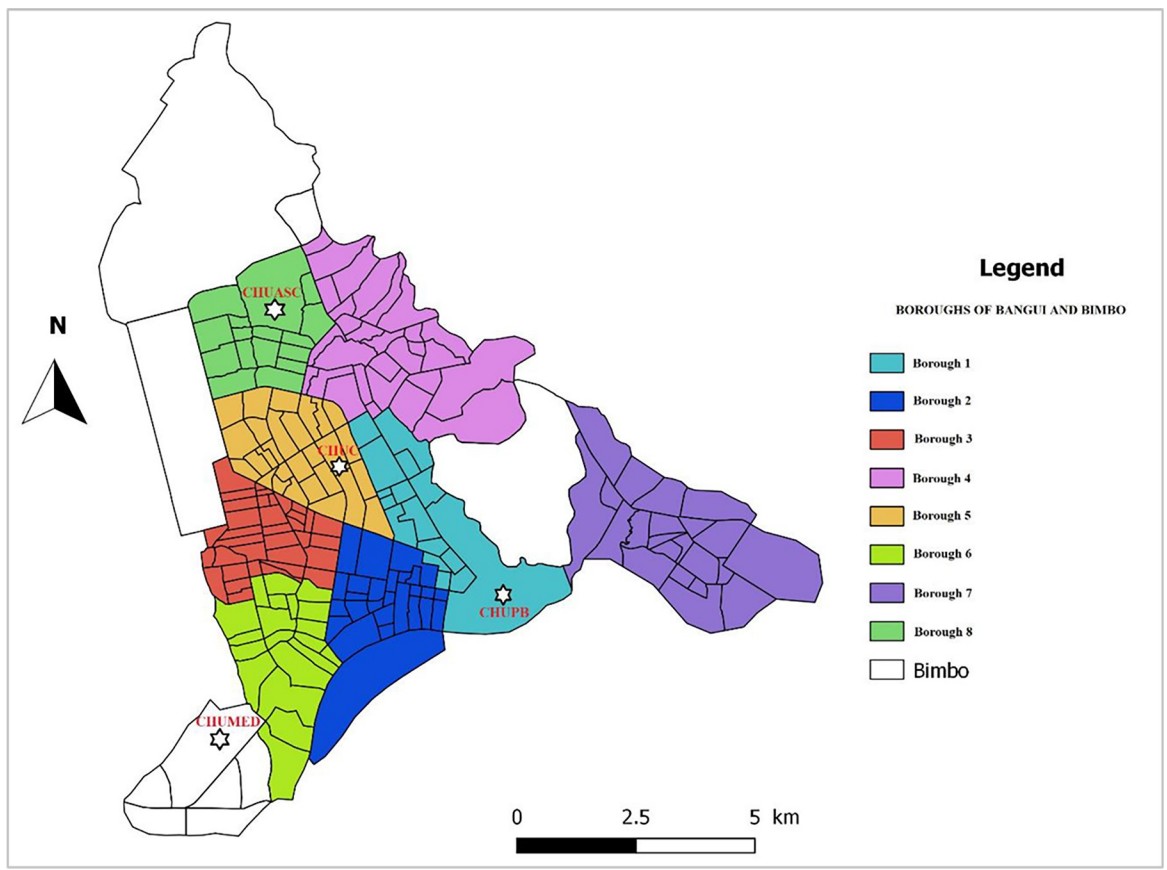

**Fig 1. Location of hospitals included in this study (this map was created with QGIS software, https://www.qgis.org/fr/site/ from the Bangui shapefile downloaded from https://data.humdata.org/dataset/cod-ab-caf).**

(CHUC) and the Centre Hospitalier Universitaire Maman Elisabeth Domitien (CHUMED) as CHUPB HCWs did not consent again for their participation in the survey.

CHUPB, CHUMED, CHUASC and CHUC are national reference teaching hospitals located in Bangui, the capital of CAR. The surface area of Bangui is 67 $km^2$ and has a population of 1,145,280, with a density of 17,094 inhabitants per $km^2$. The main missions of these hospitals are preventive and promotional care, curative care, training and research. Each of these hospitals has emergency, outpatient and inpatient departments. The number of healthcare staff are 407, 150 and 413 and 540, while their bed capacities are 296, 100, 300 and 800, respectively. Moreover, there is regular mobility of healthcare staff, particularly nurses and midwives, between these university hospitals. The Fig 1 provide the locations of these hospitals.

Our study population consisted of healthcare workers (HCWs) in hospital departments. Included were all staff administering health care to patients (medical doctors, nurses, midwives, radiology technicians, laboratory technicians and surgeons staff) and support staff (health personel having non-clinical functions) in the healthcare departments in the hospitals. Hence, this staff administering health care to patients was considered as close contact staff because they spent a lot of time with patients presenting various diseases whose infectious status to COVID-19 is unknown and most of the time they do not wear appropriate personal protective equipment.

All staff were eligible for this survey, and an informed consent form signed by each staff member was required before including them in the survey.

The protocol for this study received approval from the Ethics and Scientific Committee of the University of Bangui (N˚16/UB/FACSS/CES/20) and authorization from the Ministry of Health and Population (N˚ 934/MSP/DIRCAB/CMPSC/20) before the survey was conducted.

The sample size for the first survey was set to 179 HCWs. The sample size was calculated using OpenEpi:

(http://www.openepi.com/SampleSize/SSPropor.htm).

$$n = deff * \frac{N*p*(1-p)}{\frac{d^2}{Z}*(N-1) + p*(1-p)}$$

where N is the size of the population to be surveyed (for finite population correction factor) = 1000; p, the hypothesized (%) seroprevalence in the HCWs (15% +/−5 for the first survey and 15% +/−5 for the second survey; confidence limits as % of 100) (absolute +/−%) (d) = 5%; deff, a design effect (deff = 1); and Z, a constant (1.96 for a 95% confidence interval).

Based on the above parameters, the required sample size (n) was 122 participants for the second survey, for an expected seroprevalence of anti-SARS-CoV-2 antibodies equal to 90%, with a 95%confidence interval with a precision of 5%.

During the first and second surveys, we collected demographic data (sex and age) and qualification of the participating HCWs and blood sample of approximately 1 to 2 mL was taken in a dry tube and sent to the Institut Pasteur of Bangui. The miniVidas was used for assessing SARS-CoV-2 IgM and IgG anti-Spike (anti-S) (Spike protein by enzyme-linked fluorescent assay (ELFA). The performance of this test has been previously evaluated, as a internal control of quality at Institut Pasteur of Bangui, using 180 plasma bank samples from before May 2019, resulting in a specificity of 99·6%.

The data were analyzed using EpiInfo7 software. The link between the serological status (presence of IgG and IgM antibodies) of SARS-CoV-2 and the characteristics of the participating HCWs were investigated by comparing the percentages using the chi-square (χ2) test. In addition, odds ratios were calculated to assess the association between the individual characteristics according to IgM- and/or IgG-positive serological status regarding the first and second survey. Differences between groups were analyzed using a χ2 test at a P-values < 0,05.

## Results

A total of 179 and 141 HCWs were included in the first and second surveys, respectively. The proportions of socio-demographic categories such as age and gender, as well as the occupation were similar for the HCWs surveyed in both periods.

Given that the COVID-19 vaccine was deployed after the completion of our first survey in May 2021, during the second survey, 82.3% (116/141) of the HCWs were already vaccinated against COVID-19. Of the 116 vaccinated staff, the majority (n = 82, or 70.8%) had received the Astra Zeneca vaccine, followed by the Johnson & Johnson vaccine (n = 30, or 25.8%), and 4 (3.4%) had received a single dose of any of these two type of the COVID-19 vaccine. The proportions of close contact staff with patients (medical doctors, nurses, midwives, radiology technicians, laboratory technicians and surgeons statf) and support staff vaccinated were 83.0% (73/88) and 81.0%, respectivelly (P = 0.07). Among HCWs who received Astra Zeneca vaccine, 87.8% had 2 doses, and a 90.0% of HCWs who received Johnson & Johnson vaccine had 2 doses.

The results of the serological analysis revealed that 51/179 (31.8%) of the HCWs were positive for anti-SARS-CoV-2 IgG in the first survey (P < 0.0001), and 95.7% of the HCWs (135/141) were positive for anti-SARS-CoV-2 IgG in the second survey. In contrast, the proportion of SARS-CoV-2 IgM-positive personnel was 9.7% in the first survey, and 3.6% in the second survey (P = 0.03).

**Table 1. Characteristics of the healthcare workers surveyed for seroprevalence of anti-SARS-CoV-2 antibodies in Bangui, in April 2021 (survey 1) and May 2022 (survey 2).**

| Characteristics | Survey 1 | Survey 2 | P-value |
|---|---|---|---|
| | (N = 179) | (N = 141) | |
| | n (%) | n (%) | |
| **Gender** | | | |
| Male | 49 (27.4) | 52 (36.9) | - |
| Female | 130 (72.6) | 89 (63.1) | |
| **Mean age** (years) (±SD) | 43 (± 9.7) | 43 (± 9.2) | |
| **Age category (years)** | | | |
| <30 | 20 (11.2) | 23 (16.3) | 0.24 |
| 31–40 | 53 (29.6) | 38 (26.9) | |
| 41–50 | 57 (31.8) | 52 (36.9) | |
| > 50 | 49 (27.4) | 28 (19.9) | |
| **Hospital** | | | |
| CHUPB | 40 (22.4) | - | - |
| CHUMED | 57 (31,8) | 61 (43.3) | |
| CHUC | 82 (45.8) | 55 (24.8) | |
| CHUASC | - | 45 31.9 | |
| **HWs occupation** | | | |
| Close contacts staff with patients | 90 (50.2) | 88 (62.4) | 0.03 |
| Support staff | 89 (49.8) | 53 (37.6) | |
| **Anti-COVID-19 vaccination** | | | |
| Yes | * | 116 (82.3) | - |
| No | * | 25 (17.7) | |
| **Anti-SARS-CoV-2 IgG** | | | |
| Positive | 51 (31.8) | 135 (95.7) | <0.0001 |
| Negative | 122 (68.2) | 6 (4.3) | |
| **Anti-SARS-CoV-2 IgM** | | | |
| Positive | 17 (9.7) | 5 (3.6) | 0.03 |
| Negative | 162 (90.5) | 136 (96.4) | |

Nine HCWs have both IgG and IgM in the first survey and four HCWs have both IgG and IgM in the second one. The characteristics of these two study populations are detailed in Table 1.

The analysis of the relationship between the characteristics of the participating HCWs, neither gender (male or female), nor age categories had any effect on the positive serological status (IgM and/or IgG) in the two surveys. In the hospitals where we conducted the two surveys successively (CHUC and CHUASC), the positive serological status increased from 58.5% during the first survey to 100% during the second survey in the CHUC, whereas it increased from 77.2% to 96.7% in the CHUASC (OR, 1.83; 95% confidence interval, [1.02–3.3]; P = 0.041). Regarding the occupation of the HCWs surveyed, close contacts with patients were more likely to have a positive serological status (IgM and/or IgG) against SARS-CoV-2 compared with support staff during the ([1.3–4.3]; P = 0.005) (Table 2).

## Discussion

The results of this study on the seroprevalence of anti-SARS-CoV-2 antibodies are the first data on the serological status of COVID-19 among HCWs in the CAR. Here, we assessed this indicator over two different periods, revealing a significant increase in the proportion of HCWs with anti-COVID-19 antibodies after one year, above all, HCWs with direct contact

**Table 2. Relationship between healthcare workers and COVID-19 serological status (IgM and/or IgG) in Bangui, April 2021 (survey 1) and May 2022 (survey 2).**

| Characteristics | Survey 1 (N = 179) | | Survey 2: (N = 141) | | OR* | [95%IC] | P-value |
|---|---|---|---|---|---|---|---|
| | Positive IgM and/or IgG: n (%) | Negative IgM and/or IgG: n (%) | Positive IgM and/or IgG: n (%) | Negative IgM and/or IgG: n (%) | | | |
| **Gender** | | | | | | | |
| Male | 20 (40.8) | 29 (59.2) | 50 (96.2) | | Ref. | | |
| Female | 45 (34.6) | 85 (65.4) | 85 (95.5) | | 0.75 | [0.40–1.42] | 0.38 |
| **Age category (years)** | | | | | | | |
| ≤30 | 14 (70.0) | 6 (30.0) | 21 (91.1) | 2 (8.7) | Ref. | | |
| 31 à 40 | 31 (58,5) | 22 (41,5) | 36 (94,7) | 2 (5,3) | 0,77 | [0.34–1.77] | 0,54 |
| 41 à 50 | 38 (66,7) | 19 (33,3) | 50 (96,2) | 2 (3,8) | 1,07 | [0.48–2.42] | 0,86 |
| > 50 | 31 (63,3) | 18 (45,0) | 28 (100,0) | 0 (0,0) | 0,60 | [0.26–1.40] | 0,24 |
| **Hospital** | | | | | | | |
| CHUPB | 22 (55.0) | 18 (45.0) | - | - | NA | | |
| CHUMED | - | - | 41 (91.1) | 4 (8.9) | NA | | |
| CHUC | 49 (58.5) | 34 (41.5) | 35 (100.0) | 0 (0.0) | Ref. | | |
| CHUASC | 44 (77.2) | 13 (22.8) | 59 (96.7) | 2 (3.3) | **1.83** | [1.02–3.3] | 0.041 |
| **HCWs occupation** | | | | | | | |
| Close contact staff with patients | 27 (30.0) | 63 (70.0) | 85 (96.6) | 3 (3.3) | 2.30 | [1.3–4.3] | 0.005 |
| Support staff | 38 (42.7) | 51 (57.3) | 51 (96.2) | 2 (3.8) | | | |

*OR, odds ratio calculated by comparing IgM and/or IgG positive serological status between the first and second survey according to occupation of the HCWs. Significant values are shown in bold. CHUPB, Pediatric University Hospital of Bangui; CHUMED, Centre Hospitalier Universitaire Maman Elisabeth Domitien; CHUC, Community University Hospital; CHUASC, the University Hospital of the Sino-Central African Friendship

with patients (medical doctors, nurses, midwives, radiology technicians, laboratory technicians and surgeons statf). This is due, not only to the vaccination, but also to the continuous circulation of the virus in the general population, given that these HCWs can not only become infected in the community where they live, but they are also relatively more exposed to infection in their workplaces [9].

Hence, the synergistic effect of the COVID-19 infection and the vaccine that the staff received explains this high level of seroprevalence. Although the seroprevalence found during the first survey of this study was driven only by SARS-CoV-2 infection, the sharp increase in seroprevalence observed with the second survey indicates a complementary increase in immune status through vaccination. This sharp increase in seroprevalence may be due also to the infections caused by more transmissible variants [12].

Close contacts HCWs with patients had the highest overall seroprevalence. They were more likely to have a positive serology not only because of the anti-COVID-19 vaccination, but also because they are at high risk of being infected with SARS-CoV-2 from patients.

The relaxation of protection measures and shortage of appropriate material and protective equipment when working in the frontline or contact with patients without protection are factors related to SARS-CoV-2 infection. It has been highlighted that the effectiveness of protective measures in the prevention of SARS-CoV-2 infection is not absolute and that HCWs with the closest contact with COVID-19 patients had a more than 2-fold increase in the risk of SARS-CoV-2 infection over other HCWs [13].

Our study is in line with other studies monitoring the seroprevalence of anti-SARS-CoV-2 antibodies and the acquisition of anti-SARS-CoV-2 immunity in HCWs. Similar studies on this topic have been carried out in several countries since the emergence of the COVID-19 pandemic with varied results [14–18]. Many national and regional studies have estimated the prevalence of SARS-CoV-2 IgG antibodies in the population, and the seroprevalence in Africa (65.1% in Q3 2021) is among the highest in the world [18]. In August 2021, a sero-epidemiological survey conducted in Bangui showed that there is a high cumulative level of immunity in general population (74.1%), thus indicating a significant degree of spread of SARS-CoV-2 in the population [19]. During the same period of this survey in general population, a vaccination campaign against COVID-19 was carried out among healthcare staff. This underscores that the HCW surveyed in our study, less than one year after this campaign, have hybrid immunity (SARS-CoV-2 infection combined with anti-COVID-19 vaccination). However, the detection of anti-S antibodies did not allow us to differentiate between vaccinated individuals and others with or without a history of infection, which would have been possible if we had detected anti-nucleocapsid (anti-N) antibodies. However, it have been showed that anti-S protein and anti-N persisted up to 12 months after infection, but 97% of individuals retained their anti-S while only 20% retained their anti-N antibodies [20]. Furthermore, prior SARS-CoV-2 infection increases the titers of SARS-CoV-2 anti-S antibodies responses elicited by subsequent vaccination [21].

The IgG level in the vaccinated population can remain elevated for 25 weeks, indicating that IgG may exist for an even longer period with a positive effect against SARS-CoV-2 [22]). Our IgM and IgG antibody seropositivity rates corroborate those from other countries. For example, in a study of IgG and IgM response to SARS-CoV-2 vaccine in HCWs in China in July 2021, IgM and IgG antibody seropositivity rates were 3.1% and 74.2%, respectively [23].

The main limitation of this study is the selection bias which might happen since one of the three health facilities surveyed in the first study was not surveyed in the second study, and was replaced by another health facility. However, this bias would be minimized beacause the vaccination campaign among HCWs was carried out in all these facilities. In addition, exposure to infection would have been similar, given that the categories of close contacts staff (medical doctors, nurses, midwives, radiology technicians, laboratory technicians and surgeons statf) with patients are the same.

Another limitation of our study is that we did not evaluate antibody titers. Indeed, according to studies, IgG levels are waning rapidly by weeks after infection/vaccination or reamain at stable levels over months. hence a need to define the specific antibody titers that correlate of protection [22, 24].

## Conclusion

Our findings showed a sharp increase prevalence of IgG antibodies to SARS-CoV-2 by May 2022 compared to that of one year before. This result confirms the effect of vaccination as well as infection with SARS-CoV-2 because this virus has been circulating intensively in the general population. However, further studies are now needed on the persistence of neutralizing antibodies, the enhancement of immunogenicity against SARS-CoV-2 variants and the impact of vaccination based on timely seroprevalence data.

## Supporting information

**S1 Dataset. Anonymized survey 1 and survey 2 data bases.** This dataset can be downloaded from https://doi.org/10.6084/m9.figshare.23694741.
(XLS)

## Acknowledgments

We thank the healthcare workers who participated in this study. We express our gratitude to the hospital administration authorities for their help and advice in collecting the data. Many thanks also to Vincent Richard of the International Affairs Department at the Pasteur Institute of Paris for providing support for this study.

## Author Contributions

**Conceptualization:** Alexandre Manirakiza, Hermione Dahlia Mossoro-Kpinde, Brice Martial Yambiyo, Christian Diamant Mossoro-Kpinde, Cathy Sandra Gomelle Coti-Reckoundji, Gérard Grésenguet, Guy Vernet, Marie Astrid Vernet, Emmanuel Nakoune.

**Data curation:** Alexandre Manirakiza, Brice Martial Yambiyo, Lydie Joella-Venus de la Grace Namsenei-Dankpea, Cathy Sandra Gomelle Coti-Reckoundji, Modeste Bouhouda.

**Formal analysis:** Alexandre Manirakiza, Brice Martial Yambiyo.

**Funding acquisition:** Alexandre Manirakiza, Guy Vernet, Marie Astrid Vernet, Emmanuel Nakoune.

**Investigation:** Alexandre Manirakiza, Christian Malaka, Hermione Dahlia Mossoro-Kpinde, Christian Diamant Mossoro-Kpinde, Emmanuel Fandema, Christelle Niamathe Yakola, Rodrigue Doyama-Woza, Ida Maxime Kangale-Wando, Jess Elliot Kosh Komba, Sandra Manuella Bénedicte Nzapali Guiagassomon, Lydie Joella-Venus de la Grace Namsenei-Dankpea, Marie Astrid Vernet.

**Methodology:** Alexandre Manirakiza, Hermione Dahlia Mossoro-Kpinde, Christian Diamant Mossoro-Kpinde, Gérard Grésenguet, Guy Vernet, Marie Astrid Vernet.

**Project administration:** Alexandre Manirakiza, Guy Vernet, Marie Astrid Vernet, Emmanuel Nakoune.

**Resources:** Emmanuel Nakoune.

**Supervision:** Alexandre Manirakiza, Hermione Dahlia Mossoro-Kpinde, Christian Diamant Mossoro-Kpinde, Ida Maxime Kangale-Wando, Sandra Manuella Bénedicte Nzapali Guiagassomon, Jean-Chrisostome Gody, Marie Astrid Vernet.

**Validation:** Alexandre Manirakiza, Christian Malaka, Hermione Dahlia Mossoro-Kpinde, Brice Martial Yambiyo, Emmanuel Fandema, Christelle Niamathe Yakola, Rodrigue Doyama-Woza, Lydie Joella-Venus de la Grace Namsenei-Dankpea, Modeste Bouhouda, Jean-Chrisostome Gody, Gérard Grésenguet, Marie Astrid Vernet.

**Visualization:** Alexandre Manirakiza, Christian Malaka, Brice Martial Yambiyo, Lydie Joella-Venus de la Grace Namsenei-Dankpea, Cathy Sandra Gomelle Coti-Reckoundji, Modeste Bouhouda, Gérard Grésenguet.

**Writing – original draft:** Alexandre Manirakiza, Christian Malaka, Hermione Dahlia Mossoro-Kpinde, Brice Martial Yambiyo, Christian Diamant Mossoro-Kpinde, Emmanuel Fandema, Christelle Niamathe Yakola, Rodrigue Doyama-Woza, Jess Elliot Kosh Komba, Sandra Manuella Bénedicte Nzapali Guiagassomon, Cathy Sandra Gomelle Coti-Reckoundji, Jean-Chrisostome Gody, Gérard Grésenguet, Marie Astrid Vernet, Emmanuel Nakoune.

**Writing – review & editing:** Alexandre Manirakiza, Christian Malaka, Hermione Dahlia Mossoro-Kpinde, Brice Martial Yambiyo, Christian Diamant Mossoro-Kpinde, Emmanuel Fandema, Christelle Niamathe Yakola, Rodrigue Doyama-Woza.

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
