## [Decision Letter · Decision Letter 0]

19 May 2023

PGPH-D-22-02038

Seroprevalence of anti-SARS-CoV-2 antibodies before and after implementation of anti-COVID-19 vaccination among hospital staff in Bangui, Central African Republic

Dear Dr. Alezandre Manirakiza

Thank you for submitting your manuscript to PLOS Global Public Health. After careful consideration, we feel that it has merit but does not fully meet PLOS Global Public Health’s publication criteria as it currently stands. Therefore, we invite you to submit a revised version of the manuscript that addresses the points raised during the review process.

The reviewers have added many comments intended to improve the paper. I look forward to reading the final version as this is important work and we will hope to publish it in its future form.

Do specify which antibody tests were used. There are S (Spike protein) and N(nucleocapsid) tests; the former picks up vaccination and infection immunity, the latter only infected-based immunity.

Do we have a sense of the timing of the vaccination between the two surveys, since immunity can wane with time?

Please clarify the vaccination dosages given more clearly on lines 143-5: "*the majority (82, or 70.8%) had received the Astra Zeneca vaccine, followed by the Johnson & Johnson vaccine (30, or 25.8%), and 4 (3.4%) had received a single dose of any of these two type of the COVID-19 vaccine." *J&J is usually single dose so do clarify that 82 had 2 doses of AZ, 30 had 1 dose J&J, and 4 had one dose of a two dose vaccine, if that is correct.

We look forward to receiving your revised manuscript.

Kind regards,

Megan Coffee, MD, PhD

Academic Editor

Journal Requirements:

1. Please include the following request in the decision letter, and ping me with follow-up. “Please include a complete copy of PLOS’ questionnaire on inclusivity in global research in your revised manuscript. Our policy for research in this area aims to improve transparency in the reporting of research performed outside of researchers’ own country or community. The policy applies to researchers who have travelled to a different country to conduct research, research with Indigenous populations or their lands, and research on cultural artefacts. The questionnaire can also be requested at the journal’s discretion for any other submissions, even if these conditions are not met.  Please find more information on the policy and a link to download a blank copy of the questionnaire here: https://journals.plos.org/globalpublichealth/s/best-practices-in-research-reporting. Please upload a completed version of your questionnaire as Supporting Information when you resubmit your manuscript.”

2. Please send a completed 'Competing Interests' statement, including any COIs declared by your co-authors. If you have no competing interests to declare, please state "The authors have declared that no competing interests exist". Otherwise please declare all competing interests beginning with twhe statement "I have read the journal's policy and the authors of this manuscript have the following competing interests:"

3. Please provide separate figure files in .tif or .eps format only and remove any figures embedded in your manuscript file. Please also ensure all files are under our size limit of 10MB.

4. Please amend your Data Availability Statement and indicate where the data may be found.

Additional Editor Comments (if provided):

Reviewers' comments:

Reviewer's Responses to Questions

**Comments to the Author**

1. Does this manuscript meet PLOS Global Public Health’s publication criteria? Is the manuscript technically sound, and do the data support the conclusions? The manuscript must describe methodologically and ethically rigorous research with conclusions that are appropriately drawn based on the data presented.

Reviewer #1: Yes

Reviewer #2: Partly

Reviewer #3: Partly

2. Has the statistical analysis been performed appropriately and rigorously?

Reviewer #1: Yes

Reviewer #2: No

Reviewer #3: Yes

3. Have the authors made all data underlying the findings in their manuscript fully available (please refer to the Data Availability Statement at the start of the manuscript PDF file)?

Reviewer #1: Yes

Reviewer #2: No

Reviewer #3: No

4. Is the manuscript presented in an intelligible fashion and written in standard English?

Reviewer #1: Yes

Reviewer #2: Yes

Reviewer #3: No

5. Review Comments to the Author

Reviewer #1: The method used is acceptable in line with scientific research.

The sample size was adequate

The results and subsequent analysis was accepatable

The discussion was scientifically argued with acceptable number of references.

There is a link between, the title, discussion and conclusion.

Overall i have no objection

Reviewer #2: 1. Methods section

• Four (4) health facilities were sampled in the first and second surveys. However, two of the facilities are not consistent between the first and the second survey. This means that for the second survey, the population sampled for these two facilities is different from the first survey hence not “repeated” as the methodology mentions. It is not reported as to why there was a change and how it impacted analysis of the data.

• A brief description of the health facilities would be useful. Capacity, category (such as regional, referral, super-specialised or otherwise etc.)

• Lines 121-123: ….. The nasopharyngeal swabs were used for viral detection using RT-PCR (retro-transcriptase polymerase chain reaction) (polymerase chain reaction). Mention which virus is being referred to for viral detection, remove the second bracket and its words which are repetition and for the term “retro-transcriptase” I believe the author meant “reverse-transcriptase”.

• Occupation: In the manuscript, the breakdown of occupation is limited to medical doctors, nurses and midwives. Other cadres such as radiologists, lab, surgeons are important as well to define due to close contact with patients and samples.

• Line 125-126: To compare the two-sampling time points, the sampling should have been comparable, however, for the second survey, only blood samples were collected, while for the first survey, both blood and nasopharyngeal swabs were collected. Did the study test for positive SARS-CoV-2 samples in the second survey? At least for IgM positive?

• Which ELISA kit/ brand was used?

• Line 144-145: states that 4 (3.4%) had received a single dose of any of the two types of COVID vaccine (AstraZeneca and J&J). To note is that the J&J vaccine was a single-dose vaccine.

2. Results

• The two health facilities that were sampled only in the first survey or only in the second survey. These cannot be analysed within the scope of this manuscript, and the authors should consider omitting them from this manuscript. The author should only maintain the health facilities that were sampled at both time points.

• Table 1:

i. The ratio of male/female. The authors have included a p-value. The ratio of male/female would reflect the sex ratio of staff at their workplace therefore, no analysis between the two sexes can be done.

ii. Occupation: A more detailed breakdown is needed. Example diagnostic staff such as laboratory staff, radiologists etc. These cadres have close access to patients and samples.

iii. What was the reason for not performing a SARS-CoV-2 PCR test during the second survey? All should have been tested for the virus as per the first survey. Gives an indication of infection during the second survey, re-infection, onset and duration of IgM/IgG.

iv. A useful addition to the information in this manuscript is the antibody titres seen in both surveys and between vaccines.

v. If SARS-CoV-2 results are available for the 5 who tested IgM-positive in the second survey, this information should be in the manuscript. This may give an

vi. Did any of the HCWs have both IgG and IgM?

vii. Line 160-161: What does it mean when you say you conducted the survey successfully in two facilities i.e CHUC and CHUASC? What went wrong with the other two CHUMED and CHUPB?

viii. Which vaccine elicited the highest titre levels? AZ or J&J?

3. Discussion

• Very brief. Expand on the discussion. Discuss the results with findings from other settings.

• Discuss the significant results.

• Limitations of the study should be mentioned e.g. testing limitations that may lead to false positive or negative results. Limitations of the methodology used.

4. Conclusion

• There is a general comment in the conclusion that states that all HCWs in Bangui developed IgG antibodies by May 2021. The value of 95.7% is not 100% there are some negatives. Also, the authors did not indicate whether the facilities mentioned in Bangui are all of them.

5. References

• General comment for all references: Use the correct reference format for this journal “Vancouver”.

• Reference 12: At the end, it is stated that it was accepted for publication. If so, cite the accepted version with the DOI. I could not find it online.

• Reference 17: It is a Medriv reference. Reference the actual peer-reviewed publication.

Reviewer #3: congratulation in your study

please find some of the inputs on the research report to work on

1.please align the the report in a presentable manner where one can easily find the required information(Headings and sub- headings)

2. you can only use one format of data presentation either percentages, ratios or exact number to avoid bulk and similar data that may lead to confusion

3. provide all the required documentation in order to cross check the information

6. PLOS authors have the option to publish the peer review history of their article (what does this mean?). If published, this will include your full peer review and any attached files.

**Do you want your identity to be public for this peer review?** For information about this choice, including consent withdrawal, please see our Privacy Policy.

Reviewer #1: No

Reviewer #2: No

Reviewer #3: **Yes: **Nyakorema Lucas Ryoba

---

## [Decision Letter · Decision Letter 1]

29 Aug 2023

PGPH-D-22-02038R1

Seroprevalence of anti-SARS-CoV-2 antibodies before and after implementation of anti-COVID-19 vaccination among hospital staff in Bangui, Central African Republic

Dear Dr. Manirakiza: 

Thank you for submitting your manuscript to PLOS Global Public Health. After careful consideration, we feel that it has merit but does not fully meet PLOS Global Public Health’s publication criteria as it currently stands. Therefore, we invite you to submit a revised version of the manuscript that addresses the points raised during the review process.

The paper is almost ready for publication. Please reference the reviewer comments shared. In particular, given the difference in significance with occupation there should be more discussion and explanation as it can be confusing. originally, doctor, nurse/midwife, and other was not significant but is now that it is close contact vs support staff, there should be more discussion of this point and also more clarification of what close contact is. I would include a table of what close contacts are considered, as this is not a standardized term and the meaning may be inferred differently. In fact, radiologist may not be considered by many to have close patient contact, but a radiology tech would; however, in different settings the roles may vary and this should be clarified. Moreover, I would also provide more discussion on the N (nucleocapsid) vs S (spike) IgG testing, as here only S available, which is a limitation as cannot include the knowledge N tests would have added to the S data (ie as N positive only after infection, while S positive after both vaccination and infection).

We look forward to receiving your revised manuscript.

Kind regards,

Megan Coffee, MD, PhD

Academic Editor

Journal Requirements:

Additional Editor Comments (if provided):

Reviewers' comments:

Reviewer's Responses to Questions

**Comments to the Author**

1. If the authors have adequately addressed your comments raised in a previous round of review and you feel that this manuscript is now acceptable for publication, you may indicate that here to bypass the “Comments to the Author” section, enter your conflict of interest statement in the “Confidential to Editor” section, and submit your "Accept" recommendation.

Reviewer #2: (No Response)

Reviewer #3: All comments have been addressed

2. Does this manuscript meet PLOS Global Public Health’s publication criteria? Is the manuscript technically sound, and do the data support the conclusions? The manuscript must describe methodologically and ethically rigorous research with conclusions that are appropriately drawn based on the data presented.

Reviewer #2: Partly

Reviewer #3: Yes

3. Has the statistical analysis been performed appropriately and rigorously?

Reviewer #2: I don't know

Reviewer #3: Yes

4. Have the authors made all data underlying the findings in their manuscript fully available (please refer to the Data Availability Statement at the start of the manuscript PDF file)?

Reviewer #2: No

Reviewer #3: Yes

5. Is the manuscript presented in an intelligible fashion and written in standard English?

Reviewer #2: No

Reviewer #3: Yes

6. Review Comments to the Author

Reviewer #2: The authors have attempted to respond to all comments by the authors however there are still some clarifications that are still needed as follows:

1. General comment: Check for typographical and grammatical errors specifically for the newly added text e.g. Line 163: word “statf” change to “staff”

2. Line 94: The author has stated that CHUPS HCW did not “comply” again for their participation in the survey. I suggest that the word “comply” is changed to “consent”. I suggest that the sentence ends at the word “survey” there is no need for further explanation. I suggest removing the part that says, “due to some internal administrative concerns and some of them were transferred at CHUMED”.

3. Line 164: grammar: “Support staff was non-clinical functions” suggest to change the word “was” to “having” or another similar word.

4. Table 1: HCWs occupation: the p-value is now significant. This needs to be clarified in the rebuttal. Data from both time-points should be provided for review.

5. Discussion section: lines 7-10. Put a reference.

6. In the rebuttal, the authors have not indicated changes to the references, such as added or removed references.

Reviewer #3: Congratulations for this achievement

I can't understand why is every line numbered in your manuscript

Also you can pass through it to finalize some grammatic errors so as to provide i better research work

All the best in your publications

Thank you

7. PLOS authors have the option to publish the peer review history of their article (what does this mean?). If published, this will include your full peer review and any attached files.

**Do you want your identity to be public for this peer review?** For information about this choice, including consent withdrawal, please see our Privacy Policy.

Reviewer #2: No

Reviewer #3: No

---

## [Decision Letter · Decision Letter 2]

10 Oct 2023

Seroprevalence of anti-SARS-CoV-2 antibodies before and after implementation of anti-COVID-19 vaccination among hospital staff in Bangui, Central African Republic

PGPH-D-22-02038R2

Dear Dr Alexandre Manirakiza:

We are pleased to inform you that your manuscript 'Seroprevalence of anti-SARS-CoV-2 antibodies before and after implementation of anti-COVID-19 vaccination among hospital staff in Bangui, Central African Republic' has been provisionally accepted for publication in PLOS Global Public Health.

Best regards,

Megan Coffee, MD, PhD

Academic Editor

Reviewer Comments (if any, and for reference):

Reviewer's Responses to Questions

**Comments to the Author**

1. If the authors have adequately addressed your comments raised in a previous round of review and you feel that this manuscript is now acceptable for publication, you may indicate that here to bypass the “Comments to the Author” section, enter your conflict of interest statement in the “Confidential to Editor” section, and submit your "Accept" recommendation.

Reviewer #2: All comments have been addressed

Reviewer #4: All comments have been addressed

2. Does this manuscript meet PLOS Global Public Health’s publication criteria? Is the manuscript technically sound, and do the data support the conclusions? The manuscript must describe methodologically and ethically rigorous research with conclusions that are appropriately drawn based on the data presented.

Reviewer #2: Yes

Reviewer #4: Yes

3. Has the statistical analysis been performed appropriately and rigorously?

Reviewer #2: Yes

Reviewer #4: Yes

4. Have the authors made all data underlying the findings in their manuscript fully available (please refer to the Data Availability Statement at the start of the manuscript PDF file)?

Reviewer #2: Yes

Reviewer #4: (No Response)

5. Is the manuscript presented in an intelligible fashion and written in standard English?

Reviewer #2: Yes

Reviewer #4: Yes

6. Review Comments to the Author

Reviewer #2: (No Response)

Reviewer #4: The revised submission provides more relevant information to help further understand SARS- Cov-2 seroprevalence and clarifies the findings.

7. PLOS authors have the option to publish the peer review history of their article (what does this mean?). If published, this will include your full peer review and any attached files.

**Do you want your identity to be public for this peer review?** For information about this choice, including consent withdrawal, please see our Privacy Policy.

Reviewer #2: No

Reviewer #4: No
